

# Site-specific length-biomass relationships of arctic arthropod families are critical for accurate ecological inferences

Tom S. L. Versluijs[1,2,*], Mikhail K. Zhemchuzhnikov[1,2,*], Dmitry Kutcherov[3], Tomas Roslin[4], Niels Martin Schmidt[5,6], Jan A. van Gils[1,2] and Jeroen Reneerkens[1]

[1] Department of Coastal Systems, NIOZ Royal Netherlands Institute for Sea Research, Den Burg, Texel, The Netherlands
[2] Faculty of Science and Engineering, University of Groningen, Groningen, The Netherlands
[3] Department of Entomology, Saint Petersburg State University, Saint Petersburg, Russia
[4] Department of Ecology, Swedish University of Agricultural Sciences (SLU), Uppsala, Sweden
[5] Department of Ecoscience, Aarhus University, Roskilde, Denmark
[6] Arctic Research Centre, Aarhus University, Aarhus, Denmark
[*] These authors contributed equally to this work.

Corresponding author
Tom S. L. Versluijs,
tom.versluijs@gmail.com,
t.s.l.versluijs@rug.nl

## ABSTRACT

Arthropods play a crucial role in terrestrial ecosystems, for instance in mediating energy fluxes and in forming the food base for many organisms. To better understand their functional role in such ecosystem processes, monitoring of trends in arthropod biomass is essential. Obtaining direct measurements of the body mass of individual specimens is laborious. Therefore, these data are often indirectly acquired by utilizing allometric length-biomass relationships based on a correlative parameter, such as body length. Previous studies have often used such relationships with a low taxonomic resolution and/or small sample size and/or adopted regressions calibrated in different biomes. Despite the scientific interest in the ecology of arctic arthropods, no site-specific family-level length-biomass relationships have hitherto been published. Here we present 27 family-specific length-biomass relationships from two sites in the High Arctic: Zackenberg in northeast Greenland and Knipovich in north Taimyr, Russia. We show that length-biomass regressions from different sites within the same biome did not affect estimates of phenology but did result in substantially different estimates of arthropod biomass. Estimates of daily biomass at Zackenberg were on average 24% higher when calculated using regressions for Knipovich compared to using regressions for Zackenberg. In addition, calculations of daily arthropod biomass at Zackenberg based on order-level regressions from frequently cited studies in literature revealed overestimations of arthropod biomass ranging from 69.7% to 130% compared to estimates based on regressions for Zackenberg. Our results illustrate that the use of allometric relationships from different sites can significantly alter the biological interpretation of, for instance, the interaction between insectivorous birds and their arthropod prey. We conclude that length-biomass relationships should be locally established rather than being based on global relationships.

## INTRODUCTION

Arthropods constitute the most numerically abundant and diverse animal group in terrestrial ecosystems (*Goulson, 2019*). Global biomass of above-ground arthropods is estimated to be similar to the biomass of all humans and their livestock combined (*Rosenberg et al., 2023*). Across the globe, arthropods play essential roles in nutrient cycling (*Yang & Gratton, 2014*), and in food webs, where they serve as *e.g.*, pollinators and/or as both prey and predators (*Ollerton, Winfree & Tarrant, 2011*; *Schmidt et al., 2017*). For example, ca. 60% of all bird species are insectivorous and rely on arthropods as a resource for survival, growth and egg production (*Morse, 1971*; *Klaassen et al., 2001*; *Piersma et al., 2003*), and 88% of all plant species are estimated to depend on animal pollination, most of which can be attributed to insects (*Ollerton, Winfree & Tarrant, 2011*; *Goulson, 2019*).

Due to the integral role of arthropods in mediating ecosystem functions, long-term ecological monitoring of trends in arthropod diversity, abundance and biomass is essential (*Hallmann et al., 2017*; *Goulson, 2019*; *Gillespie et al., 2020*). Data on arthropod biomass are important in ecological studies, for instance in characterizing size—abundance relationships (*White et al., 2007*) and in measuring energy fluxes among habitats and/or within food webs (*Polis & Hurd, 1995*). Moreover, detailed information on arthropod biomass is key to understanding whether and when the temporal asynchrony between the breeding phenology of insectivorous birds and arthropod availability translates into fitness consequences (*Durant et al., 2005*; *Ramakers, Gienapp & Visser, 2019*). Data on biomass can also provide a better understanding of processes underlying changes in arthropod community structure (*Southwood, Moran & Kennedy, 1982*; *Robertson et al., 2012*), because biomass scales with metabolic rate and thus serves as an indicator of the functional role of a species within the arthropod community (*Gillooly et al., 2001*; *Saint-Germain et al., 2007*).

The importance of data on arthropod biomass is ubiquitous, but acquiring body mass measures for each individual arthropod specimen is laborious. A less time-consuming alternative is to derive estimates of body mass from a correlative parameter such as body length (*e.g.*, *Rogers, Buschbom & Watson, 1977*; *Sample et al., 1993*). This requires knowledge of the allometric relationship between body length and body mass for individual prey taxa. Such relationships typically follow a power function (*e.g.*, *Gowing & Recher, 1984*; *Hayes & Shonkwiler, 2006*) and are frequently used to estimate arthropod biomass (*e.g.*, *Saint-Germain et al., 2007*; *McKinnon et al., 2012*; *Lister & Garcia, 2018*).

Despite the prevalent use of such allometric relationships, they have several limitations. In particular, four types of extrapolations can reflect into biased inferences regarding arthropod biomass: (I) Empirically quantified allometric relationships are generally restricted to temperate regions (*Rogers, Buschbom & Watson, 1977*; *Schoener, 1980*; *Sample et al., 1993*; *Sohlström et al., 2018*), the subtropics (*Sage, 1982*) or the tropics (*Schoener, 1980*; *Ganihar, 1997*; *Gruner, 2003*; *Sohlström et al., 2018*), while detailed regressions for other regions—such as the Arctic—are lacking. As arthropods may have specific morphological adaptations to their local environment (*e.g.*, *Strathdee & Bale, 1998*), applying allometric relations parameterized for one region to another may result in biased estimates of arthropod biomass (*Schoener, 1980*; *Hodar, 1996*; *Baumgärtner & Rothhaupt,*

*2003*, but see *Gowing & Recher, 1984*). (II) Empirically quantified allometric relationships are seldom available at a family level or lower taxonomical levels (but see *e.g.*, *Sample et al., 1993*). As a result, order-level taxonomical equations are frequently used to estimate biomass (*e.g.*, *Sage, 1982*; *Senner, Stager & Sandercock, 2017*; *Sohlström et al., 2018*). Resorting to such coarse taxonomic resolution may be problematic because length-biomass relationships can vary remarkably even within the lower taxonomical levels (*Johnston & Cunjak, 1999*; *Baumgärtner & Rothhaupt, 2003*). (III) Empirically quantified allometric relationships are generally based on datasets with limited sample sizes (*e.g.*, *Hodar, 1996*; *Sabo, Bastow & Power, 2002*). (IV) Empirically quantified allometric relationships are often based on data from several decades ago (*e.g.*, *Rogers, Buschbom & Watson, 1977*), while the morphology of arthropods may have changed over time (*Bowden et al., 2015*; *Polidori et al., 2020*; *Wonglersak et al., 2021*).

In this study, we present allometric length-biomass relationships at high (family-level) taxonomic resolution from two sites in the High Arctic. Drawing on these detailed, site-specific data, we show that estimates of daily arthropod biomass can differ substantially when calculated using length-biomass relationships parameterized for different sites within the same biome, or when they are based on order-level regressions extracted from frequently cited studies from other biomes. Our results demonstrate the importance of using site-specific length-biomass relationships at high taxonomic resolution to improve the accuracy of biomass estimates and enhance biological inferences.

## MATERIALS & METHODS

### Allometric length-biomass regressions for Zackenberg and Knipovich
*Data collection and processing*

To derive allometric length-biomass relationships for arctic arthropods at high taxonomical resolution (family level), and to compare the generality of such relations between areas, we used data from two high arctic sites. Arthropods were caught using yellow pitfalls in June–August 2015 in Zackenberg, northeast Greenland (74°28′N, 20°34′W, $N = 3,594$) and June–July 2018 in Knipovich, Taimyr, Russia (76°04′N, 98°32′E, $N = 799$). Upon collection, specimens were stored in 96% ethanol and later identified to family level, except for Collembola and Acari which were identified to sub-class level. The length of all specimens was measured under a stereomicroscope to the nearest 0.1 mm directly after taking them out of the ethanol preservative. Lengths were measured from the frons to the tip of the abdomen, excluding any appendages such as antennas, proboscis, or ovipositor. Once measured, all specimens were dried for 2–4 days in open air until their biomass remained constant. All specimens were subsequently oven-dried for 20–24 h at 60 °C, after which they were placed in a desiccator filled with silica gel to prevent increases in biomass due to moisture absorption. The dry mass of all specimens was weighed directly after taking them out of the desiccator on a microscale balance with an accuracy of 0.01 mg. In general, we aimed to determine the dry mass for each individual specimen, but, to reduce the relative effect of measurement error (*Mährlein et al., 2016*), we grouped specimens that were too small to be weighed individually into several length classes and subsequently calculated an average dry mass per length class.

*Fitting statistical models*

We fit separate length-biomass regressions for all taxonomic groups in Zackenberg and Knipovich, and fit separate models for data measured at the level of individuals and for data averaged per length class. We only fit models when at least eight specimens or groups were measured. We fit four linear models per taxonomic group: (I) an intercept-only model: $W = B0$, (II) a linear model on untransformed data: $W = B0 + B1*L$, (III) a linear model on natural-log transformed data, *i.e.*, an exponential model: $\ln(W) = B0 + B1*L$, and (IV) a linear model on natural log–log transformed data, *i.e.*, a power model: $\ln(W) = B0 + B1*\ln(L)$, where W = dry mass, L = body length, B0 corresponds to the intercept and B1 to the slope of the linear model. We then selected the best model for each taxonomic group based on AIC (*Burnham & Anderson, 2002*). For all models we visually checked normality assumptions using QQ-plots and checked homoscedasticity assumptions by plotting standardized Pearson residuals against fitted values and against body length (*Zuur et al., 2009*). To quantify uncertainty for the fitted allometric equations, we calculated 95% quantile confidence intervals for model predictions and regression coefficients using non-parametric (case) bootstrapping using 10,000 bootstrap samples (*Efron & Tibshirani, 1994*; *Nakagawa & Cuthill, 2007*). We only used bootstrapping for taxa with a sample size of at least 20. We corrected body mass predictions from log-linear models using Duan's smearing factor (*Duan, 1983*; *Mährlein et al., 2016*). For the full derivation of the appropriate model, dealing with outliers and quantification of model uncertainty, see Article S1.

## Allometric length-biomass regressions extracted from literature

To compare our regressions for Zackenberg and Knipovich to those often used in literature, we selected nine frequently cited studies containing allometric length-biomass relationships of terrestrial arthropods: *Rogers, Buschbom & Watson (1977)*; *Schoener (1980)*; *Sage (1982)*; *Gowing & Recher (1984)*; *Sample et al. (1993)*; *Hodar (1996)*; *Ganihar (1997)*; *Sabo, Bastow & Power (2002)*; *Gruner (2003)*. As these studies only contained family-level regressions for Hymenoptera *Ichneumonidae*, but not for the other arthropod families in our dataset (Table S1), we extracted regressions at order-level taxonomic resolution. We then selected the three most cited studies (extracted from Web of Science on 6 July 2023) that provided regressions for all three taxonomic orders in our dataset (*i.e.*, Araneae, Diptera and Hymenoptera).

## Estimates of arthropod biomass and phenology based on different allometric regressions

To establish how our perception of seasonal trajectories in arthropod biomass would differ depending on the origin of the length-biomass regressions employed, we derived estimates of (I) average daily arthropod biomass, and (II) the timing of the median date of arthropod biomass, for 24 years of arthropod data collected at Zackenberg when biomass was inferred using family-level length–biomass regressions for either Zackenberg, Knipovich, or order-level length-biomass regressions from literature.

### Arthropod data

We analyzed 24 years of arthropod data collected at Zackenberg between 1996 and 2019 (*Greenland Ecosystem Monitoring, 2020*). These data were not part of the data used to derive allometric regressions for Zackenberg, as the latter was based on additional arthropod data collected in 2015. Sampling has occurred at near-weekly intervals from the moment of snowmelt until late August or late September (*Schmidt et al., 2016*). Arthropods were trapped using yellow pitfall traps at six plots with dimensions $10 \times 20$ m$^2$ (*Schmidt et al., 2016*). One plot was not operational between 1999 and 2018 and was therefore excluded from our analysis. To prevent biases due to interannual differences in the duration of the trapping window, we restricted our analysis to a fixed period from day of year 157 (5-6 June) to 238 (25-26 August). All collected specimens were identified at family-level taxonomic resolution, except for Acari and Collembola which were identified to sub-class level. We restricted our analysis to the taxonomic groups: Araneae *Linyphiidae*, Diptera *Chironomidae*, Dip. *Empididae*, Dip. *Muscidae*, Dip. *Mycetophilidae*, Dip. *Sciaridae*, and Hymenoptera *Ichneumonidae*, because these were the only families for which we were able to calculate length-biomass regressions for both Zackenberg and Knipovich (excluding Collembola as they made a very limited contribution to overall biomass). This subset included 234,487 specimens, corresponding to 30.1% of the total number of specimens for all taxonomic groups, which corresponds to 25.0% of total biomass.

### Estimating arthropod biomass

The selected Zackenberg arthropod data contain counts of specimens per taxonomic group with a timestamp corresponding to the date when a trap is emptied. These counts thus reflect the cumulative number of specimens collected during all the days for which a trap was active. We first translated this into daily counts per taxonomic group by calculating the average number of trapped specimens per taxonomic group for each day a trap was active. To infer seasonal trajectories in biomass from these count data, we then allocated a length to each specimen by random sampling from taxon-specific length distributions (for more details see Article S1). Once a length was allocated to each individual, we used our taxon-specific length-biomass regressions to calculate its corresponding biomass. This latter step was carried out five times, utilizing the regressions specific to Zackenberg, Knipovich, and those extracted from the three studies selected from literature. For each of these five biomass variables, we then calculated the average arthropod biomass per trap per day for each year. In addition, we estimated arthropod phenology for each year by calculating the date when 50% of cumulative biomass was reached (hereafter "median date of arthropod biomass") using linear interpolation. We obtained 95% quantile confidence intervals for all estimated parameters using non-parametric (case) bootstrapping with 10,000 bootstrap samples (*Efron & Tibshirani, 1994*; *Nakagawa & Cuthill, 2007*). All statistical analyses were performed in R version 4.1.2 (*R Core Team, 2021*).

Portions of this text were previously published as part of a preprint (https://www.biorxiv.org/content/10.1101/2023.04.04.534924v1.full).

## RESULTS

### Allometric length-biomass regressions

We identified 4,389 arthropod specimens belonging to 42 taxonomic groups (Zackenberg, $n = 3,590$ specimens of 34 taxonomic groups, and Knipovich, $n = 799$ of 19 taxonomic groups). Body length was measured for 4,383 individual specimens, while biomass was measured for 1,573 individual specimens. The remaining 2,785 individuals, for which biomass could not be individually determined, were grouped into length classes for which an average length and biomass was calculated per group. For 27 taxonomic groups, sample size was sufficiently large to construct allometric relationships for one or both sites, resulting in 22 regressions for Zackenberg and 15 regressions for Knipovich (Table 1, Table S2). We fit two regressions for Acari and two for Diptera *Chironomidae* at Zackenberg (*i.e.*, one for data measured at the level of individuals and one for data averaged per length class). For 31 out of 37 allometric relationships the best supported statistical model was a linear model fitted on natural log–log transformed data, *i.e.*, a power model (Table S3). The allometric relationships of the eight arthropod taxa for which data were available for both Zackenberg and Knipovich are shown in Fig. 1, while the relationships for arthropod taxa for which data were only available for either site are shown in Figs. S1 and S2, respectively. The average calculated smearing factor across all length-biomass regressions was 1.041 [95% CI, 1.027, 1.058]. Since we used a natural-log transformed response variable, body mass predictions on the arithmetic scale would thus underestimate arthropod biomass by 4.1% (and as much as 25.4% for Ichneumonidae) unless this correction was made (Table 1).

From the nine studies selected from literature, *Rogers, Buschbom & Watson (1977)*, *Hodar (1996)* and *Ganihar (1997)* were the three most cited studies that provided allometric regressions for all three arthropod orders in our analysis (*i.e.*, Aranea, Diptera and Hymenoptera, Table S1). The extracted allometric relationships from literature are summarized in Table 2.

### Estimates of arthropod biomass and phenology based on different allometric regressions

Estimates of the average arthropod biomass per trap per day at Zackenberg were on average 23.9% [95% CI: 23.5, 24.4] higher when calculated using regressions for Knipovich than when calculated using regressions for Zackenberg (Fig. 2, Table 3). When biomass was calculated using order-level regressions extracted from literature, estimates of the average arthropod biomass per trap per day at Zackenberg were between 69.7% and 129.7% higher than when biomass was calculated using regressions for Zackenberg (Fig. 2, Table 3). Taxon-specific differences between our family-level regressions from Zackenberg and Knipovich, and the order-level regressions extracted from literature are visualized in Figure S3.

Estimates of the median date of arthropod biomass at Zackenberg were on average 0.13 days [95% CI: 0.03, 0.26] earlier when regressions for Knipovich were used instead of regressions for Zackenberg (Table 3). In addition, when arthropod biomass was calculated using order-level regressions extracted from literature, the median date of arthropod biomass occurred on average between 0.09 and 0.78 days earlier than when regressions for
**Table 1 Best supported allometric length-biomass relationships for 27 arthropod taxa for Zackenberg ($n = 22$) and Knipovich ($n = 15$).** The column 'n' depicts the number of data points on which each allometric model was fit with, if applicable, in brackets the sample size before averaging within different length classes. 'Min (mm)' and 'Max (mm)' indicate the minimum and maximum of length ranges of specimens used to fit each regression. 'SF' depicts the smearing factor used to correct backtransformed predictions for models with a natural-log transformed response variable. 'Level' indicates whether the allometric model was fit on individual level measurements or average values for different length classes. 'Location' indicates the site where the specimens were collected. Case-bootstrapped 95% confidence intervals for all model parameters can be found in Table S2.

| Taxon | n | Min (mm) | Max (mm) | B0 | B1 | SF | Level | Model | Location |
|---|---|---|---|---|---|---|---|---|---|
| Aca *sp.* | 6 (605) | 0.29 | 1.64 | −3.627 | 2.012 | 1.028 | avg | ln(W/SF) = B0 + B1 * ln(L) | ZAC |
| Aca *sp.* | 9 | 2.08 | 3.24 | −3.438 | 3.249 | 1.008 | ind | ln(W/SF) = B0 + B1 * ln(L) | ZAC |
| Ara *Dictynidae* | 8 | 2.06 | 2.50 | −5.903 | 5.646 | 1.052 | ind | ln(W/SF) = B0 + B1 * ln(L) | ZAC |
| Ara *Linyphiidae* | 28 | 2.30 | 3.70 | −2.422 | 1.928 | 1.013 | ind | ln(W/SF) = B0 + B1 * ln(L) | KNP |
| Ara *Linyphiidae* | 25 | 0.69 | 2.62 | −3.556 | 2.938 | 1.035 | ind | ln(W/SF) = B0 + B1 * ln(L) | ZAC |
| Ara *Lycosidae* | 129 | 1.74 | 8.68 | −3.718 | 2.931 | 1.013 | ind | ln(W/SF) = B0 + B1 * ln(L) | ZAC |
| Ara *Thomisidae* | 12 | 2.55 | 5.55 | −3.268 | 2.963 | 1.007 | ind | ln(W/SF) = B0 + B1 * ln(L) | ZAC |
| Clm *sp.* | 9 (209) | 0.65 | 2.56 | −0.015 | 0.026 | NA | avg | W = B0 + B1 * L | KNP |
| Clm *sp.* | 8 (1002) | 0.20 | 1.71 | −5.129 | 1.196 | 1.008 | avg | ln(W/SF) = B0 + B1 * ln(L) | ZAC |
| Col *Carabidae* | 21 | 6.30 | 8.30 | 5.371 | NA | NA | ind | W = B0 | KNP |
| Col *Chrysomelidae* | 34 | 4.90 | 6.60 | −3.672 | 3.384 | 1.011 | ind | ln(W/SF) = B0 + B1 * ln(L) | KNP |
| Col *Staphylinidae* | 56 | 2.90 | 7.50 | −4.403 | 2.498 | 1.051 | ind | ln(W/SF) = B0 + B1 * ln(L) | KNP |
| Dip *Anthomyiidae* | 30 | 2.74 | 7.41 | −2.648 | 0.510 | 1.039 | ind | ln(W/SF) = B0 + B1 * L | ZAC |
| Dip *Bolitophilidae* | 16 | 3.30 | 4.80 | −4.061 | 1.559 | 1.050 | ind | ln(W/SF) = B0 + B1 * ln(L) | KNP |
| Dip *Ceratopogonidae* | 37 (337) | 1.42 | 2.78 | −3.476 | 0.765 | 1.019 | avg | ln(W/SF) = B0 + B1 * ln(L) | ZAC |
| Dip *Chironomidae* | 83 | 1.16 | 6.10 | −4.738 | 2.414 | 1.150 | ind | ln(W/SF) = B0 + B1 * ln(L) | KNP |
| Dip *Chironomidae* | 67 (328) | 1.31 | 2.93 | −0.028 | 0.031 | NA | avg | W = B0 + B1 * L | ZAC |
| Dip *Chironomidae* | 45 | 2.70 | 7.64 | −6.286 | 3.127 | 1.058 | ind | ln(W/SF) = B0 + B1 * ln(L) | ZAC |
| Dip *Culicidae* | 78 | 3.97 | 6.94 | −3.093 | 1.313 | 1.098 | ind | ln(W/SF) = B0 + B1 * ln(L) | ZAC |
| Dip *Empididae* | 30 | 5.90 | 7.80 | −2.151 | 1.467 | 1.011 | ind | ln(W/SF) = B0 + B1 * ln(L) | KNP |
| Dip *Empididae* | 22 | 3.57 | 8.50 | −4.295 | 2.350 | 1.025 | ind | ln(W/SF) = B0 + B1 * ln(L) | ZAC |
| Dip *Muscidae* | 44 | 4.00 | 7.80 | −4.685 | 2.949 | 1.013 | ind | ln(W/SF) = B0 + B1 * ln(L) | KNP |
| Dip *Muscidae* | 412 | 3.84 | 8.80 | −4.679 | 2.835 | 1.025 | ind | ln(W/SF) = B0 + B1 * ln(L) | ZAC |
| Dip *Mycetophilidae* | 32 | 3.40 | 5.40 | −3.597 | 1.923 | 1.031 | ind | ln(W/SF) = B0 + B1 * ln(L) | KNP |
| Dip *Mycetophilidae* | 21 | 3.72 | 5.45 | −5.411 | 2.701 | 1.022 | ind | ln(W/SF) = B0 + B1 * ln(L) | ZAC |
| Dip *Phoridae* | 14 | 1.71 | 3.28 | −0.116 | 0.091 | NA | ind | W = B0 + B1 * L | ZAC |
| Dip *Scathophagidae* | 35 | 5.45 | 8.93 | −3.717 | 2.370 | 1.022 | ind | ln(W/SF) = B0 + B1 * ln(L) | ZAC |
| Dip *Sciaridae* | 110 | 1.56 | 3.50 | −3.854 | 0.525 | 1.047 | ind | ln(W/SF) = B0 + B1 * L | KNP |
| Dip *Sciaridae* | 48 (273) | 1.62 | 3.75 | −5.164 | 2.384 | 1.030 | avg | ln(W/SF) = B0 + B1 * ln(L) | ZAC |
| Dip *Syrphidae* | 9 | 6.58 | 12.62 | −5.314 | 3.087 | 1.012 | ind | ln(W/SF) = B0 + B1 * ln(L) | ZAC |
| Dip *Tachinidae* | 11 (28) | 10.56 | 12.48 | −3.952 | 2.617 | 1.005 | avg | ln(W/SF) = B0 + B1 * ln(L) | ZAC |
| Dip *Tipulidae* | 52 | 9.70 | 15.80 | −3.033 | 1.945 | 1.024 | ind | ln(W/SF) = B0 + B1 * ln(L) | KNP |
| Dip *Trichoceridae* | 31 | 3.10 | 5.90 | −6.197 | 3.234 | 1.035 | ind | ln(W/SF) = B0 + B1 * ln(L) | KNP |
| Hym *Ichneumonidae* | 22 | 2.40 | 6.70 | −5.739 | 3.226 | 1.063 | ind | ln(W/SF) = B0 + B1 * ln(L) | KNP |

**Table 1** (*continued*)

| Taxon | n | Min (mm) | Max (mm) | B0 | B1 | SF | Level | Model | Location |
|---|---|---|---|---|---|---|---|---|---|
| Hym *Ichneumonidae* | 50 | 1.86 | 12.34 | −5.559 | 2.928 | 1.254 | ind | ln(W/SF) = B0 + B1 * ln(L) | ZAC |
| Hym *Tenthredinidae* | 12 | 4.30 | 8.10 | −5.434 | 3.261 | 1.061 | ind | ln(W/SF) = B0 + B1 * ln(L) | KNP |
| Lep *Nymphalidae* | 61 | 12.34 | 15.09 | 3.408 | −0.330 | 1.023 | ind | ln(W/SF) = B0 + B1 * ln(L) | ZAC |

**Notes.**

Abbreviations: Aca, Acari; Ara, Araneae; Clm, Collembola; Col, Coleoptera; Dip, Diptera; Hym, Hymenoptera; Lep, Lepidoptera; ind, individual level weight measurements; avg, averaged weight estimates per length class; W, body mass (mg); L, body length (mm); KNP, Knipovich; ZAC, Zackenberg.

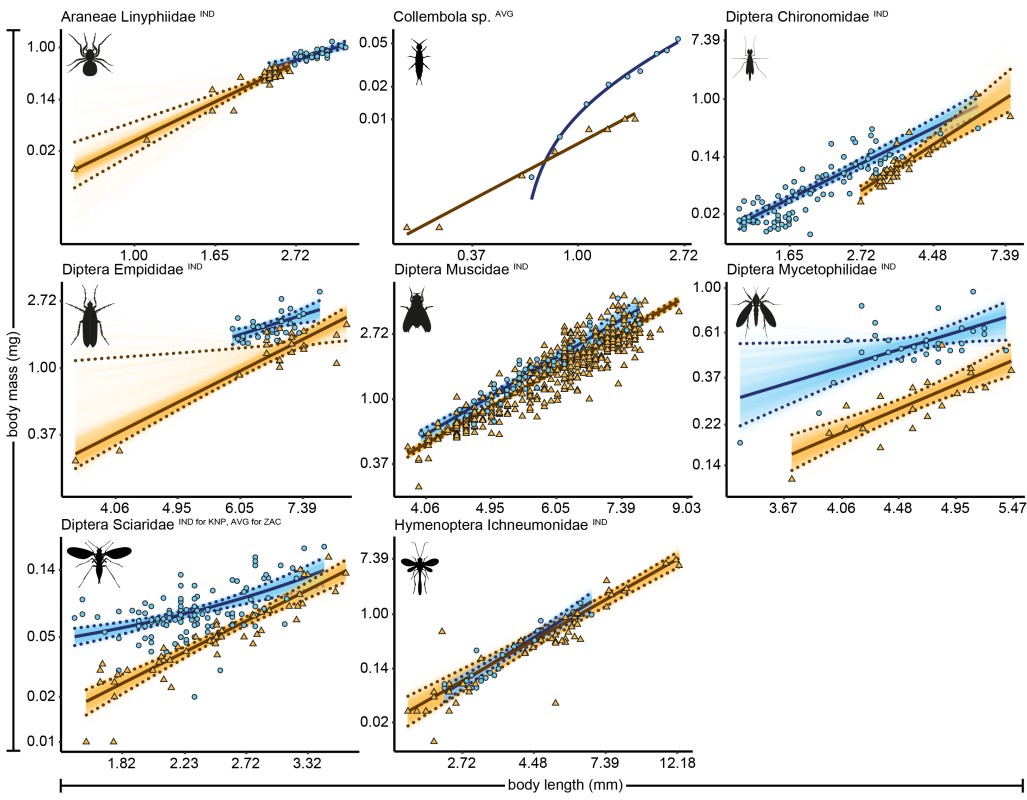

**Figure 1** **Length-biomass allometric relationships for eight arthropod taxa for which data were available for both Zackenberg and Knipovich.** Data for Zackenberg (ZAC) are depicted as orange triangles and data for Knipovich (KNP) as blue circles. Axes are log-transformed but labelled with non-transformed values. Superscripts following the taxonomic names indicate whether datapoints represent individual level weight measurements ('IND') or averages per length class ('AVG'). Solid lines indicate the best supported model for each taxon. Dotted lines indicate 95% quantile confidence intervals calculated over 10,000 case bootstrapping runs. Individual bootstrapping runs are drawn as transparent lines to create a colour gradient that visualizes the distribution of best fitting models over all bootstrapping runs. Wide confidence intervals (*e.g.*, for *Empididae*) are an artefact of the use of case-resampling in combination with influential datapoints.

Zackenberg were used (Table 3). The maximum difference in estimated median date of arthropod biomass for a single year occurred in 2015 where this metric was 5.6 days earlier when based on *Rogers, Buschbom & Watson (1977)* instead of based on regressions from Zackenberg (Table 3).

**Table 2  Overview of order-level regressions extracted from literature.** The column 'n' depicts the number of data points on which each allometric model was fit. 'Min (mm)' and 'Max (mm)' indicate the minimum and maximum of length ranges of specimens used to fit each regression. 'B0' and 'B1' correspond to the parameters of the allometric regression depicted under 'Model'. 'Location' indicates the site where the specimens were collected.

| Taxon | n | Min (mm) | Max (mm) | B0 | B1 | Model | Location | Reference |
|-------|---|----------|----------|-----|-----|-------|----------|-----------|
| *Ara sp.* | 114 | 1.0 | 12.7 | −3.211 | 2.468 | $\ln(W) = B0 + B1 * \ln(L)$ | IN_Goa | *Ganihar (1997)* |
| *Dip sp.* | 20 | 1.8 | 16.0 | −3.429 | 2.594 | $\ln(W) = B0 + B1 * \ln(L)$ | IN_Goa | *Ganihar (1997)* |
| *Hym sp.* | 26 | 2.4 | 10.0 | −3.592 | 2.643 | $\ln(W) = B0 + B1 * \ln(L)$ | IN_Goa | *Ganihar (1997)* |
| *Ara sp.* | 18 | 1.3 | 27.1 | −2.260 | 2.296 | $\ln(W) = B0 + B1 * \ln(L)$ | ES_GRX | *Hodar (1996)* |
| *Dip sp.* | 36 | 1.0 | 24.0 | −3.467 | 2.392 | $\ln(W) = B0 + B1 * \ln(L)$ | ES_GRX | *Hodar (1996)* |
| *Hym sp.* | 24 | 1.6 | 26.5 | −1.810 | 1.900 | $\ln(W) = B0 + B1 * \ln(L)$ | ES_GRX | *Hodar (1996)* |
| *Ara sp.* | 25 | 0.7 | 12.0 | −3.106 | 2.929 | $\ln(W) = B0 + B1 * \ln(L)$ | USA_WA | *Rogers, Buschbom & Watson (1977)* |
| *Dip sp.* | 84 | 0.9 | 34.0 | −3.293 | 2.366 | $\ln(W) = B0 + B1 * \ln(L)$ | USA_WA | *Rogers, Buschbom & Watson (1977)* |
| Hym sp. | 97 | 0.7 | 27.0 | −3.871 | 2.407 | $\ln(W) = B0 + B1 * \ln(L)$ | USA_WA | *Rogers, Buschbom & Watson (1977)* |

**Notes.**

Abbreviations:: Ara, Araneae; Dip, Diptera; Hym, Hymenoptera; W, body mass (mg); L, body length (mm); IN_Goa, Goa India; ES_GRX, Granada Spain; USA_WA, Washington state United States of America.

## DISCUSSION

Based on 24 years of data, we show how the use of family-level allometric relationships from two sites within the same (arctic) biome can result in substantially different estimates of daily arthropod biomass, despite employing identical methodology and taxonomic resolution. In addition, estimates of daily arthropod biomass calculated using order-level regressions extracted from literature were considerably larger than estimates based on family-level regressions for Zackenberg. This corroborates the findings of earlier studies showing distinct variation within taxonomic groups in regression coefficients or in estimated biomass of invertebrates among sites and/or habitats (*Schoener, 1980*; *Hodar, 1996*; *Sabo, Bastow & Power, 2002*; *Baumgärtner & Rothhaupt, 2003*, but see *Gowing & Recher, 1984*). Although we show that estimates of arthropod phenology based on different regressions are on average unimportant from a biological perspective (*i.e.*, smaller than a day), estimates for a single year could differ up to 5.6 days.

Inconsistencies in family-level regression coefficients, or biomass estimates, across sites may arise from differences in site-specific species compositions, or because variation in the time, location and type of sampling (*e.g.*, yellow pitfalls versus sweep netting) may yield different subsets of sampled species when species differ in their phenology and/or small-scale spatial distribution (*Høye & Forchhammer, 2008*). In addition, variation in regression parameters might occur due to differences in the timing of emergence among dimorphic sexes (*Danks & Oliver, 1972*; *McLachlan, 1986*), or because species differ in their morphological adaptations to their local environment (*Schoener, 1980*; *Strathdee & Bale, 1998*). For instance, arthropods in the tropics have been suggested to have longer and thinner bodies than those in temperate areas (*Schoener, 1980*; *Sohlström et al., 2018*). Differences in habitat characteristics and/or food availability may also affect regression parameters by causing intraspecific variation in growth rates (*Griffith, Perry & Perry, 1993*; *Johnston & Cunjak, 1999*). Although we employed identical methodologies for Zackenberg

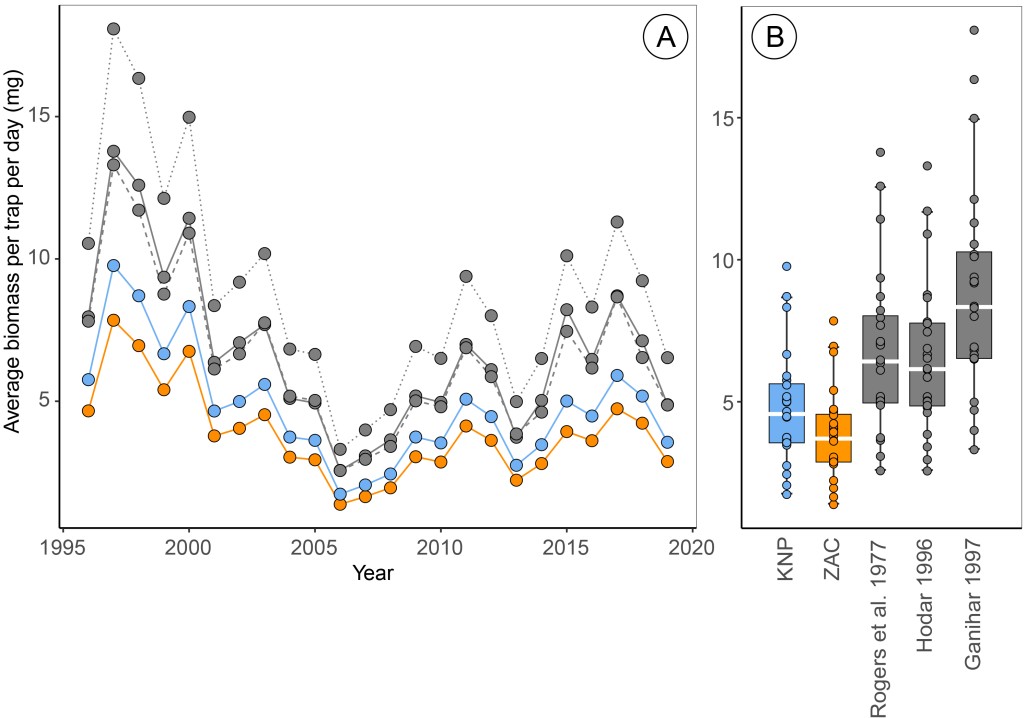

**Figure 2** **Estimates of average biomass per pitfall trap per day at Zackenberg (1996–2019), calculated based on regressions from five different sources.** Data include the Araneae family *Linyphiidae*, Diptera families *Chironomidae*, *Empididae*, *Muscidae*, *Mycetophilidae* and *Sciaridae* and Hymenoptera family *Ichneumonidae*. Data depicted in blue are calculated using family-level length-biomass regressions for Knipenberg (KNP) and data in orange using family-level regressions for Zackenberg (ZAC). Data depicted in grey are calculated using order-level regressions extracted from literature, where the solid grey line is based on regressions from *Rogers, Buschbom & Watson (1977)*, the dashed grey lines on regressions from *Hodar (1996)* and the dotted grey line on regressions from *Ganihar (1997)*. Boxplots summarize the spread in the data, where horizontal white bars indicate the median, the box depicts the interquartile range and whiskers represent 1.5 times the interquartile range from the upper/lower quartile.

**Table 3** **Differences in the estimated median date of arthropod biomass and the average arthropod biomass per trap per day at Zackenberg.** All depicted differences are relative to biomass estimates calculated based on family-level regressions for Zackenberg. The column 'Regressions' indicates which length-biomass regressions were used to calculate biomass. The column 'Mean diff. peak (days)' indicates the mean difference in estimated median date of arthropod biomass (in days) relative to when Zackenberg regressions were used to calculate this metric, while the column 'Max diff. peak (days)' indicates the maximum difference for this metric (in days) for a single year. The column 'Mean diff. biomass (%)' indicates the mean difference in the average arthropod biomass per trap per day (as a percentage) relative to when Zackenberg regression were used to calculate this metric, while the column 'Max diff. biomass (percentage)' indicates the maximum difference for this metric for a single year. Values in square brackets include 95% quantile confidence intervals calculated using non-parametric case-bootstrapping.

| Regressions | Mean diff. peak (days) | Max diff. peak (days) | Mean diff. biomass (%) | Max diff. biomass (%) |
|---|---|---|---|---|
| Knipovich (this study) | 0.13 [0.03, 0.26] | 1.44 | 23.9 [23.5, 24.4] | 27.3 |
| *Ganihar (1997)* | 0.30 [0.01, 0.73] | 4.29 | 129.7 [126.6, 133.4] | 157.0 |
| *Hodar (1996)* | 0.09 [−0.39, 0.72] | 5.56 | 69.7 [66.7, 73.0] | 89.6 |
| *Rogers, Buschbom & Watson (1977)* | 0.78 [0.41, 1.31] | 5.63 | 75.6 [72.3, 79.8] | 108.9 |

and Knipovich, comparisons of regression parameters or biomass estimates among studies might be hampered by differences in methodologies used for the measuring and weighing of arthropods (*Schoener, 1980*; *Johnston & Cunjak, 1999*; *Méthot et al., 2012*). For instance, corrections for back-transformations from the logarithmic to the arithmetic scale are frequently overlooked (*e.g.*, *Rogers, Buschbom & Watson, 1977*; *Sohlström et al., 2018*). Variation in body width of specimens might also explain some of the variation between regression parameters (*Sohlström et al., 2018*), although this may only lead to marginal improvements for allometric relationships constructed at the family level (*Sample et al., 1993*; *Gruner, 2003*).

Our results highlight that applying length-biomass relationships calibrated for one site to another site could result in significantly biased estimates of arthropod biomass, even when compared at family-level resolution within the same biome. Accurate estimates of arthropod biomass are essential to understanding food web dynamics and processes driving community structure (*e.g.*, *Saint-Germain et al., 2007*) and are for instance crucial in calculating the minimum amount of arthropod biomass required to sustain average growth and survival of birds (*Schekkerman et al., 2003*; *Saalfeld et al., 2019*). In addition, the use of length-biomass regressions from different sites can affect estimates of the relative contribution of different prey taxa to total prey biomass (*Hodar, 1996*), impacting estimates of prey availability for insectivores. We hope that our family-level length-biomass regressions for the Arctic will result in more accurate estimates of, for instance, global above-ground arthropod biomass (*Rosenberg et al., 2023*), local population trends in (arctic) arthropod biomass (*Andersson et al., 2022*), and prey availability for insectivorous birds (*Reneerkens et al., 2016*; *Zhemchuzhnikov et al., 2021*). Because our family-level length-biomass equations differ considerably from those from non-arctic regions and from those constructed at order-level taxonomic resolution, we argue that site-specific equations with high taxonomical resolution will provide the most accurate description of local trends in arthropod biomass and will lead to the most accurate biological inference.

## CONCLUSIONS

We hypothesized that estimates of arthropod biomass in the Arctic were biased by the use of old allometric relationships from other regions and/or by low taxonomical resolution. While the use of allometric relationships from different sites—even within the same biome—and from lower-taxonomical studies in different biomes had limited effect on estimates of arthropod phenology, they did drastically affect estimates of arthropod biomass. As such, this can affect biological interpretations regarding ecological relationships, such as the balance between trophic layers and the food available for offspring growth. Ideally, future studies should establish arthropod length-biomass relationships based on local samples and with high taxonomical resolution.

## ACKNOWLEDGEMENTS

We thank Aarhus University for providing logistics at Zackenberg. We are grateful for the assistance of many co-workers in the field, and in particular Jannik Hansen and Lars H. Hansen (Zackenberg). Zdenek Gavor and Elin Jørgensen, Aarhus University, sorted and identified arthropods collected in Zackenberg. Arthropods collected in Knipovich were measured and weighed at the Chromas core facility, St. Petersburg State University Research Park. Data from the Greenland Ecosystem Monitoring Programme were provided by the Department of Ecoscience, Aarhus University, Denmark in collaboration with Greenland Institute of Natural Resources, Nuuk, Greenland, and Department of Biology, University of Copenhagen, Denmark.

### Funding

This work was supported by the Netherlands Organisation for Scientific Research (NWO) with an Open grant (ALWOP.432) to Jan van Gils and Jeroen Reneerkens, and by a Polar Program grant (ALWPP.2016.044) and a Vici grant to Jan A. van Gils (VI.C.182.060). Additional support for Jeroen Reneerkens came from the Metawad project awarded by Waddenfonds (WF209925) and by an International Polar Year grant from NWO (886.15.207). Tomas Roslin was funded by the Academy of Finland (VEGA, grant 322266) and by the European Research Council (ERC) under the European Union's Horizon 2020 research and innovation programme (ERC-synergy grant 856506—LIFEPLAN). There was no additional external funding received for this study. The funders had no role in study design, data collection and analysis, decision to publish, or preparation of the manuscript.

### Grant Disclosures

The following grant information was disclosed by the authors:
Netherlands Organisation for Scientific Research open grant (NWO): ALWOP.432.
Netherlands Organization for Scientific Research (NWO) Polar Program grant: ALWPP.2016.044.
Netherlands Organization for Scientific Research (NWO) Vici grant: VI.C.182.060.
Netherlands Organization for Scientific Research (NWO) International Polar Year grant: 886.15.207.
The Metawad project awarded by Waddenfonds: WF209925.
VEGA grant awarded by the Academy of Finland: 322266.
ERC-synergy grant awarded by The European Research Council (ERC) under the European Union's Horizon 2020 research and innovation programme: 856506 - LIFEPLAN.

### Competing Interests

The authors declare that there are no competing interests.

## Author Contributions

- Tom S.L. Versluijs conceived and designed the experiments, performed the experiments, analyzed the data, prepared figures and/or tables, authored or reviewed drafts of the article, and approved the final draft.
- Mikhail K. Zhemchuzhnikov conceived and designed the experiments, performed the experiments, analyzed the data, prepared figures and/or tables, authored or reviewed drafts of the article, and approved the final draft.
- Dmitry Kutcherov performed the experiments, authored or reviewed drafts of the article, and approved the final draft.
- Tomas Roslin conceived and designed the experiments, authored or reviewed drafts of the article, and approved the final draft.
- Niels Martin Schmidt conceived and designed the experiments, performed the experiments, authored or reviewed drafts of the article, and approved the final draft.
- Jan A. van Gils conceived and designed the experiments, authored or reviewed drafts of the article, and approved the final draft.
- Jeroen Reneerkens conceived and designed the experiments, performed the experiments, authored or reviewed drafts of the article, and approved the final draft.

## Data Deposition

The raw data and R-code are available at Zenodo: Versluijs, Tom Sebastiaan Laurens, Zhemchuzhnikov, Mikhail K., Kutcherov, Dmitry, Roslin, Tomas, Martin Schmidt, Niels, van Gils, Jan A., & Reneerkens, Jeroen. (2023). Site-specific length-biomass relationships of arctic arthropod families are critical for accurate ecological inferences. https://doi.org/10.5281/zenodo.8124306.

## Supplemental Information

Supplemental information for this article can be found online at http://dx.doi.org/10.7717/peerj.15943#supplemental-information.

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
