# Peer review of "Site-specific length-biomass relationships of arctic arthropod families are critical for accurate ecological inferences"

_PeerJ, doi:10.7717/peerj.15943_

## Round 0.1 · original submission · Minor Revisions

Please make all amendments according to the requirements.

Reviewer 1 ·

Basic reporting

1. Here the authors present a well-written and carefully carried out study. There is a need for the information presented in the manuscript, since relying on allometric length-biomass relationships is standard practice in many ecological studies yet is prone to problems that the authors point out in the introduction. Having a published guide for biomass of arthropods found at Zackenberg is in and of itself a valuable resource due to the long-term ecological research that takes place there. In addition to these data, the authors present a well-considered analysis of potential biases of using allometric length-biomass relationships that are not site-specific.

2. In the introduction and discussion, I would like to see a bit more about potential uses for arthropod biomass data outside of use by birds as a food resource. While I know that trophic mismatches with migratory birds is a major research focus in the Arctic, the biomass of arthropods is used in many other ways - for example, trophic structures among arthropod guilds, relationships with plant species and flower phenology, the global decline in insect abundance, responses to changes in snowmelt and extreme temperatures, etc. I think the authors could do a bit more to show the relevance of the paper to non-avian ecologists.

3. Lines 242-253 This paragraph accurately describes the uncertainties to be considered when using prey biomass calculations to understand consumption, but it is not a strong ending to the paper. I suggest condensing this paragraph and adding more to the discussion about some of the interesting patterns in the results. For example, why might Zackenberg arthropods have less mass/unit length than Knipovich across multiple families? What conclusions might be drawn when arthropod biomass is estimated to be higher or lower than it actually is?

4. Minor comments: Line 120 ‘fit’ not ‘fitted’

Lines 117-120 This was a bit unclear to me, perhaps start by describing that models were fit separately for individuals vs. weight classes and then add the caveat that models were only fit when at least 8 specimens or groups were measured.

Experimental design

1. The authors should consider adding to the analysis shown in Figure 2 by making use of ‘standard’ length-biomass regressions (e.g. from Rogers et al. 1977 or Sample et al. 1993 – perhaps chose whatever is most often cited). Given how the introduction was written, I was at first expecting that there would be a comparison of site-specific and ‘standard’ published data. One would expect that using ‘standard’ data would be even more biased than using a similarly performed analysis at another Arctic research site, but it would be interesting to compare. Knowing the ways in which that bias trends and how much for each family would be useful in interpreting other Arctic studies that have made use of such ‘standard’ allometric relationships.
2. Since these estimates are based on yellow pitfall trap data, somewhere the authors should mention that these allometric relationships would not necessarily be suitable for studies using other trapping types (e.g. sweep-net).

Validity of the findings

Minor comments only:

Lines 121-123 Often using such different transformations on the same data means that one or more is going to fail model assumptions of normality & heteroscedasticity, but I don’t see any mention here or in the supplement of checking model assumptions. The authors may very well have done these checks, but it should at least be mentioned in the supplement.

Lines 149-152 I believe there needs to be stronger justification for the choice of these taxonomic groups. Are these the only overlapping families of the 22 and 15 described in line 183? If that is the case, perhaps it would be better to simply state ‘we then analyzed all groups for which we were able to calculate regressions at both sites’ and then list which groups met your criteria in the results.

Lines 152-153 Do the authors have any data to estimate the % contribution of biomass of these groups? As the authors have already pointed out, biomass is potentially more meaningful than abundance when understanding the overall importance of arthropod families.

Lines 181-183 If there are 27 taxonomic groups, 22 of which are at Zackenberg and 15 of which are at Knipovich, there should be 10 groups that were found at both sites. Figure 1 shows only 8. After looking at Table 1 and the supplement, I believe this is a typo and line 181 should say 29 instead of 27.

Additional comments

Supporting Information:
1. In the last sentence of the ‘model selection’ paragraph, please add a citation or more justification for preferring power-models.

2. ‘Calculating biomass’ section and Table 1 – why are there two relationships for Chironomidae at Zackenberg? I don’t see anything mentioned about this in the methods or results.

·

Basic reporting

In their manuscript, the authors investigated whether differences in the regressions between body length and biomass of arthropods calculated at the family level at two different sites affect the estimation of arthropod biomass for one of the sites. The manuscript is very well written, in clear and concise language. It is also very well structured and hence easy to follow. It does contain sufficient background and context to emphasize both the relevance of the study for ecological research. The results are presented with an appropriate number of tables and figures, both are very well designed and structured. Overall, this is an excellent manuscript. The authors have also provided all necessary data and Rcodes which are very thoroughly documented and commented, which makes it very easy for readers to follow and repeat the analysis. I want to applaud the authors for the care and effort they put into the documentation of the data and analysis as this is unfortunately not common.

Experimental design

The aim and content of the manuscript are within the aims and scope of the journal. The research question is well defined and very relevant both for studies on insectivorous birds (the context in which the authors placed their study question) and other studies working with arthropod abundance or biomass.
The field work was conducted following standard methodology and the authors generated a large dataset which is more than sufficient to answer the study question. The measurement of arthropod biomass has been conducted very carefully in order to ensure measurement of dry weight consistently across taxonomic groups. The methods are described in enough detail that they can easily be followed, including the statistical analyses.

Validity of the findings

All underlying data have been provided; they are robust, statistically sound, & controlled. The conclusion drawn from the data are valid and supported by the results.

Additional comments

I cannot make any recommendation to further improve the manuscript regarding its content, structure, approach, or language. However, I want to suggest adding one or two additional sentences in the methods (or maybe even in the introduction) on the three different datasets used and how they were used in the estimation of arthropod biomass. In particular, it wasn’t immediately clear to me that two different datasets from Zackenberg were used and that the dataset over 24 years was used to apply the regression models which were calculated based on the two other datasets.

---

## Round 0.2 · accepted · Accept

I am satisfied with the amendments made by authors.